# Membrane Separation of Gaseous Hydrocarbons by Semicrystalline Multiblock Copolymers: Role of Cohesive Energy Density and Crystallites of the Polyether Block

**DOI:** 10.3390/polym13234181

**Published:** 2021-11-29

**Authors:** Md. Mushfequr Rahman

**Affiliations:** Helmholtz-Zentrum Hereon, Institute of Membrane Research, Max-Planck-Straße 1, 21502 Geesthacht, Germany; mushfequr.rahman@hereon.de; Tel.: +49-415-287-2224

**Keywords:** copolymer, membrane, hydrocarbon, cohesive energy density, gas separation, semicrystalline polymer

## Abstract

The energy-efficient separation of hydrocarbons is critically important for petrochemical industries. As polymeric membranes are ideal candidates for such separation, it is essential to explore the fundamental relationships between the hydrocarbon permeation mechanism and the physical properties of the polymers. In this study, the permeation mechanisms of methane, ethane, ethene, propane, propene and n-butane through three commercial multiblock copolymers PEBAX 2533, PolyActive1500PEGT77PBT23 and PolyActive4000PEGT77PBT23 are thoroughly investigated at 33 °C. This study aims to investigate the influence of cohesive energy density and crystallites of the polyether block of multiblock copolymers on hydrocarbon separation. The hydrocarbon separation behavior of the polymers is explained based on the solution–diffusion model, which is commonly accepted for gas permeation through nonporous polymeric membrane materials.

## 1. Introduction

Multiblock copolymers, which have alternating series of polyether-based soft blocks and a glassy or semicrystalline hard block (e.g., polyamide [1,2], polyester [3,4], polyimide [5] and polyurethane [6]), have earned a reputation as ideal membrane materials for gas separation. The structure–property relationships of this class of polymers have been an intriguing field of research in the last decade. A good microphase separation between the hard and soft blocks is highly desirable to enhance the gas separation performance of these polymers [7,8]. Gas permeation occurs through the phase composed of the soft polyether blocks while the hard blocks provide mechanical strength and a film-forming ability [4,9]. Since the gas permeation behavior of such polymers can be tuned by choosing the type, content and length of the hard and soft blocks, these multiblock copolymers are often regarded as a versatile tool in the design of gas separation membranes [7,10,11,12,13,14,15,16]. These types of multiblock copolymers have been extensively explored for the separation of carbon dioxide from light gases, e.g., nitrogen and hydrogen [17,18,19,20]. The quadrupolar moment of carbon dioxide has a distinct affinity towards the polar ether oxygens, which makes them ideal membrane materials through which carbon dioxide permeates faster than the light gases. Polymers with several polyethers (e.g., poly(ethylene oxide) (PEO) [15,21], poly(propylene oxide) [22,23,24] and poly(tetramethylene oxide) (PTMO) [25,26,27]) as soft blocks have been explored as gas separation membrane materials. Multiblock copolymers with PEO as a soft block have higher carbon dioxide selectivity than others due to higher ether oxygen content. For the separation of carbon dioxide from light gases, the PEO blocks are expected to be in an amorphous state under these operating conditions because the presence of PEO crystallites reduces the gas permeability substantially. Gas permeation through non-porous polymers occurs due to the presence of fractional free volume (i.e., the volume unoccupied by the polymer chains). Gases cannot permeate through the perfectly packed chain-folded crystallites because they are too dense. In other words, the PEO crystallites do not have the free-volume to allow permeation of the gases, which lowers the gas permeability. The crystallinity of the multiblock copolymers has a relatively small influence on selectivity. The overall selectivity of the polymers is mostly dictated by the amorphous phase. However, in some studies, it has been reported that the crystallites contribute to the polymer’s size-sieving ability and, thereby, have a negative impact on the selectivity of carbon dioxide over light gases [15]. Unlike carbon dioxide separation, the hydrocarbon separation mechanism of these multiblock copolymers is relatively less explored. The separation of hydrocarbons is crucial in the petrochemical industry because it is related to obtaining high-quality fuel and raw materials for bulk chemical production (e.g., ethene for polyethylene) [28]. The conventional separation techniques (e.g., cryogenic distillation) have a large energy penalty compared to membrane separation technology. Thus, the fundamental knowledge regarding the correlation between the separation mechanism of hydrocarbons with the physical properties of the polymeric membrane materials is crucial. In this work, we have investigated the permeability, diffusivity and solubility of methane, ethane, ethene, propane, propene and n-butane through three multiblock copolymers. The three commercially available multiblock copolymers, PEBAX 2533, PolyActive1500PEGT77PBT23 and PolyActive4000PEGT77PBT23, are denoted as P2533, P1500 and P4000, respectively. The chemical structure of these polymers is provided in Figure 1. P2533 is composed of 80 wt% PTMO and 20 wt% polyamide 12 [1,2,18]. P1500 and P4000 are composed of 77 wt% poly(ethylene glycol)terephthalate and 23% poly(butylene terephthalate) [3,17]. P1500 and P4000 contain PEO segments of 1500 and 4000 g/mol, respectively. A thorough investigation of these three polymers revealed important information regarding the role of cohesive energy density and the crystallites of the polyether blocks on the gas permeation mechanism.

## 2. Materials and Methods

PolyActive was purchased from PolyVation. PEBAX 2533 was purchased from ARKEMA. The solvents dichloromethane (purity 99.0%) and n-butanol (purity 99.5%) were purchased from Merck KGaA and Scharlau Chemie S.A., respectively. The chemicals were used as received without any further purification. 

Dense films were prepared in Teflon molds via solution casting. P2533 was dissolved in n-butanol at 70 °C for 2 h. P1500 and P4000 were dissolved in dichloromethane at room temperature. The obtained homogeneous solutions were poured in a Teflon mold. The solution of P1500 and P4000 was evaporated at room temperature, while the solution of P2533 was evaporated at 40 °C. The films were dried under a vacuum overnight at 30 °C. Membrane thickness was measured by a digital micrometer, and they varied from 100 to 300 μm. 

Differential scanning calorimetry (DSC) was used to study the melting transitions of P2533, P1500 and P4000, ranging from −100 °C to 250 °C. All DSC runs were performed in a DSC 1 (Star system) from Mettler Toledo using a nitrogen purge gas stream (60 mL/min) at a scan rate of 10 K/min. Heating and cooling scans were performed by initially heating the sample to 100 °C to erase the effects of residual solvent, and then the sample was cooled to −100 °C. Finally, a second heating scan was performed up to 250 °C. The DSC thermograms presented in Figure 1 correspond to the second heating cycle.

Single gas permeability of the prepared dense membranes was determined by the constant volume and variable pressure (“time-lag”). Permeability (P), diffusivity (D) and solubility (S) are determined at 33 °C from the pressure increase curves obtained during the “time-lag” experiments using the following equations:(1)$$P=D.S=\frac{V_{p}.l.}{A.R.T.\Delta t}ln\frac{p_{f}-p_{p1}}{p_{f}-p_{p2}}$$
(2)$$D=\frac{l^{2}}{\theta }$$
where *V_p_* is the permeate volume, l is the membrane thickness, A is the membrane area, R is the gas constant, *p_f_* is the feed pressure considered constant in the time range Δ*t*, *p_p1_* and *p_p2_* are permeate pressures at times 1 and 2, Δ*t* is the time difference between two points (1 and 2) on the pressure curve and *θ* is the time lag.

The ideal selectivity of the membranes is determined according to the following equation:(3)$$\alpha_{A/B}=\frac{P_{A}}{P_{B}}=\frac{D_{A}}{D_{B}}.\frac{S_{A}}{S_{B}}$$
where *α_A/B_* is the ideal selectivity, and *P_A_* and *P_B_* are single gas permeabilities of the two gases A and B, respectively.

## 3. Results and Discussion

Figure 1 shows the second heating traces of the DSC thermograms of P2533, P1500 and P4000. The microphase separated multiblock copolymer P2533 has two distinct melting endotherms for the PTMO and polyamide 12 block, respectively. Two separate melting endotherms are also visible for P1500 and P4000 for the PEO and poly(butylene terephthalate), respectively [17]. The onset and endset of the melting endotherms of the PEO block of P1500 and P4000 are significantly different from each other. For P4000, the melting of the PEO block starts at 40 °C and ends at 49 °C. The gas permeation properties through the polymers are investigated at 33 °C. From the DSC thermograms, it is evident that at 33 °C, the polyether blocks of P2533 and P1500 are completely amorphous while that of P4000 is semicrystalline. Since the content of the polyether blocks P2533, P1500 and P4000 are rather similar, this study sheds light on the permeation mechanism of the hydrocarbons through the amorphous PTMO block, amorphous PEO block and semicrystalline PEO block, respectively. The permeabilities of all the hydrocarbons through P2533 > P1500 > P4000 are presented in Figure 2. The semicrystalline nature of the PEO blocks is responsible for the low permeability of the gases through P4000. The densely packed crystallites are impermeable for hydrocarbons. Gases permeate through the amorphous part of the polymer only. The substantially high gas permeability through P2533 compared to P1500, despite the similar amorphous polyether content, stems from the difference in the cohesive energy density of the polyether blocks of these two polymers. For all of these polymers, permeabilities of paraffinic hydrocarbons (i.e., methane, ethane, propane and butane) increase with the molecular size (Figure 2). Despite the smaller size, the permeabilities of ethene are slightly higher than that of ethane. Similarly, the permeabilities of propene are higher than propane. To get a clear understanding of the gas permeation behavior of these polymers, it is important to examine the influence of the cohesive energy density and crystallinity of the polyether blocks on the solubilities and diffusivities of the gases. 

By definition, cohesive energy density is the internal energy of a substance per unit volume, measuring the interaction energy between the molecules of the substance at a fixed temperature: the stronger the interaction between the molecules, the higher the cohesive energy density. The square root of the cohesive energy density is the Hildebrand solubility parameter. For small molecules, cohesive energy density can be determined experimentally from the enthalpy of vaporization [29]. As the polymers degrade before vaporization, the cohesive energy density and the Hildebrand solubility parameter for polymers are usually theoretically determined by a group contribution method [30]. Efforts have been made to extract the cohesive energy density from other parameters, e.g., surface tension and thermal expansivity [31]. The cohesive energy density of PEO and PTMO are widely reported in the literature. While there are inherent limitations in determining the absolute values, it is common knowledge that PEO has a higher cohesive energy than PTMO [31,32]. The attractive forces between the polyether segments stem from polar ether oxygen. The higher ether oxygen content of PEO leads to a higher cohesive energy density compared to that of PTMO.

A combination of two thermodynamic processes is involved in the dissolution of a gas molecule in a polymeric membrane—the condensation of the gas molecules at the surface of the membrane and the formation of a molecular scale gap in the polymer to accommodate the gas molecule [33]. The first process is dictated by the inherent condensability of a gas molecule. The critical temperature of the gas molecule is widely used as a measure of condensability. Therefore, in Figure 3, the solubilities of the hydrocarbons in P2533, P1500 and P4000 are plotted against their critical temperature. The second process of gas dissolution is related to the cohesive energy density of the polymer. The higher the cohesive energy density, the higher the energy demand to open up a molecular scale gap to accommodate the gas molecule at the surface of the membrane. Since the cohesive energy density of the PTMO segments is significantly lower than the PEO segments, the PTMO containing P2533 can accommodate the condensed gas molecules more easily than the PEO containing P1500 and P4000. Therefore, the solubilities of the gases are higher in P2533 compared to P1500 and P4000. From Figure 3, it is clear that the cohesive energy densities of the polymer segments have a stronger influence on the solubility of a gas with low condensability, e.g., methane. For a gas with low condensability, the solubility is mostly determined by the energy required to form a molecular-scale gap in the polymer for a gas molecule. As the condensability of the gas increases, this process starts to become less dominant when determining the solubility. In P2533, the solubilities of the gases increase systematically with the critical temperature, i.e., the condensability of the gases. However, in P1500 and P4000, the solubilities of the hydrocarbons are not dependent only on the condensability of the gases. While the solubility of paraffinic hydrocarbons increases linearly, the olefinic hydrocarbons (i.e., ethene and propene) have higher solubility than expected from their condensability. Considering the condensability of the gases, ethene is expected to have a slightly lower solubility in the polymers than ethane, and propene is expected to have a slightly lower solubility than propane. Figure 3 shows that in P1500 and P4000, the solubility of ethene is slightly higher than ethane while that of propene is significantly higher than propane. Hence, the olefinic hydrocarbons have a specific affinity towards the polymers, which the paraffinic hydrocarbons do not have. The specific affinity originates from the polar ether oxygen of the polymers and the double bonded carbons of the gases [34]. Since both PTMO and PEO contain ether oxygen, this observation leads to a conjecture that the specific affinity between polyether and olefinic gases impacts the solubility selectivity of the gases only when the cohesive energy between the polyether segments is high enough. 

The diffusion of a gas molecule through the polymeric membranes consists of a series of diffusive jumps. In a rubbery polymer, the diffusive jumps are facilitated by the formation of transient free volumes. The segments of the rubbery polymers have sufficient energy for chain rotation, translational motion and vibrational motion, which creates transient free volumes and allows the gas molecules to jump from one site to another. Hence, it is intuitive that cohesive energy density, combined with other factors, influences the formation of the transient free volume in rubbery polymers. Due to the influence of other factors (e.g., the chemical structure of the polymer), a high cohesive energy density does not necessarily translate in low fractional free volume [29]. To examine diffusion, it is equally important to consider the properties of gas molecules because the diffusive jumps are a function of the size of the gases. There are several scaling parameters for the size of a gas molecule, e.g., kinetic diameter and critical volume. Since most of the hydrocarbons are non-spherical, critical volume is a more accurate scaling parameter than kinetic diameter to compare the diffusion of the hydrocarbons [35,36,37]. For this reason, in Figure 4, the diffusivities are plotted against the critical volumes of the hydrocarbons (the critical volume values reported by Li et al. [38] are used). The diffusivities of the hydrocarbons decrease with the increasing critical volume of the gases in P1500 and P4000, which is not the case in P2533. The diffusion of the gases through the PTMO containing P2533 does not vary significantly upon the change of the hydrocarbons’ size. Hence, the higher cohesive energy density of the PEO segments leads to the size-sieving ability of P1500 and P4000. 

The influence of the crystallites’ presence on gas permeation behavior is often explained using a two-phase model originally proposed by Michaels et al. [39,40,41] for polyethylene. The model assumes that due to the presence of impermeable crystallites, the gas molecules have to follow a rather tortuous path. Moreover, the crystallites also reduce the mobility of the neighboring amorphous segments. Thus, along with the content of the amorphous fraction, it is necessary to consider a tortuosity factor, *τ*, and a chain immobilization factor, *β*, to explain the gas permeation through semicrystalline polymers. According to this model
(4)$$P_{s}=\frac{P_{a}}{\tau \beta }\emptyset_{a}$$
(5)$$S_{s}=S_{a}\emptyset_{a}$$
(6)$$D_{s}=\frac{D_{a}}{\tau \beta }$$
where *ϕ_a_* is the volume fraction of the amorphous phase in the polymer, *P_s_* is the permeability of a gas through a semicrystalline polymer, *P_a_* is the permeability of a gas through a pure amorphous polymer, *S_s_* is the solubility coefficient of a gas in a semicrystalline polymer, *S_a_* is the solubility coefficient of a gas in a pure amorphous polymer, *D_s_* is the diffusion coefficient of a gas through a semicrystalline polymer, and *D_a_* is the diffusion coefficient of the gas through a completely amorphous polymer. P4000 has lower solubilities due to the lower amorphous PEO content than P1500 (Figure 3). The diffusivities of the hydrocarbons through P4000 are also lower (Figure 4) than those through P1500. However, both polymers show a similar trend of change in the diffusivities and solubilities of the hydrocarbons as a function of critical volume and temperature, respectively. An accurate analysis of the impact of cohesive energy density and PEO crystallites is possible by comparing the permselectivities, diffusion selectivities and solubility selectivities of the gas pairs. 

Figure 5, Figure 6 and Figure 7 show the contribution of the diffusion and solubility selectivities to the permselectivities of paraffinic hydrocarbon pairs. All three polymers selectively permeate the larger paraffinic hydrocarbons of the gas pairs due to the dominant solubility selectivity over diffusion selectivity. For P2533, the diffusion selectivity of the paraffinic gas pairs is in the range of 0.9–1.5. Since the diffusion selectivities are almost equal to one, P2533 has almost no size-sieving ability, and the permselectivity is merely determined by the solubility selectivity. However, that is not the case for P1500 and P4000. Both of these polymers have substantially stronger solubility selectivities for the paraffinic hydrocarbon pairs than those of P2533. For the gas pairs with significant differences, e.g., n-butane/methane (Figure 6c) and propane/methane (Figure 6f), the solubility selectivities of P1500 and P4000 are 6–8 times higher than those of P2533. However, owing to the cohesive energy density of the PEO blocks, since P1500 and P4000 have size-sieving abilities (i.e., smaller gases diffuse faster), the diffusion selectivity values are far below one. The counteracting influence of the faster diffusivities of smaller gases and the higher solubilities of larger gases in P1500 and P4000 reduces the permselectivities of the paraffinic hydrocarbon pairs compared to those of P2533. The permselectivities of n-butane over ethane (Figure 5d) and methane (Figure 6a) are slightly higher for P4000 than those for P1500, which implies that the presence of PEO crystallites leads to slightly higher permselectivities of these two gas pairs, originating from solubility selectivities. The difference in the permselectivities and solubility selectivities of P1500 and P4000 for the other paraffinic and olefinic/paraffinic hydrocarbon pairs are relatively insignificant. The PEO crystallites significantly reduce the available surface area for the dissolution of the gases. The significantly higher condensability of n-butane over ethane and methane translates into a slight increase in the solubility selectivity in the presence of the PEO crystallites. However, the trend of solubilities (Figure 3) and the difference in solubility selectivities of all hydrocarbon pairs (Figure 5, Figure 6, Figure 7 and Figure 8) in P1500 and P4000 imply that the characteristic thermodynamic properties of the amorphous PEO of these two polymers are similar. The PEO crystallites of P4000 do not impose sufficient stress to alter the thermodynamic property of the amorphous PEO. The similar trend of diffusivities (Figure 4) and the similar diffusion selectivities of all hydrocarbon pairs (Figure 5, Figure 6, Figure 7 and Figure 8) in P1500 and P4000 prove that the PEO crystallites do not alter the size-sieving characteristics of the amorphous PEO part of P4000. From an energy consideration, the chain immobilization factor, *β* (Equation (6)), is a function of the size of permeating gases. The similar size-sieving ability of P1500 and P4000 implies that the chain immobilization factor, *β* (Equation (6)), does not play a significant role in the diffusion of hydrocarbons through P4000. The intercrystalline spaces of P4000 are sufficiently larger than the critical volume of n-butane (i.e., the largest hydrocarbon used in this study), which makes *β* relatively insignificant for the gases’ diffusion. The crystallites of PEO contribute to the tortuosity factor, *τ* (Equation (6)), only. As the hydrocarbons have to follow a long, tortuous path due to the presence of the PEO crystallites, the diffusivities through P4000 are lower than those through P1500.

## 4. Conclusions

The detailed investigation of gas permeation through the three commercial multiblock copolymers P2533, P1500 and P4000 demonstrates that for the separation of paraffinic hydrocarbons, the multiblock copolymers with PTMO segments are better membrane materials compared to those containing PEO segments. The lower cohesive energy density of the PTMO containing polymer leads to higher permeability and permselectivity of the paraffinic hydrocarbons. Due to the lower cohesive energy density, the PTMO containing polymer facilitates the dissolution of the hydrocarbons, while the diffusion of the hydrocarbons through the polymer is almost independent of the size of the gases. Under these circumstances, the solubility selectivity determines the overall permselectivity for the PTMO containing polymer. The higher cohesive energy density not only lowers the permeability of hydrocarbons but also imparts a size-sieving property in the PEO containing polymers. The size-sieving property counteracts the solubility selectivity and lowers the permselectivity of the PEO-containing polymers. However, the PEO containing polymers are an ideal choice for olefin/paraffin separation. The PEO-containing polymers allow the selective permeation of olefins over paraffins due to higher solubility, which stems from the specific affinity of the polar ether oxygen towards the double bonds of olefins. The polymer’s size-sieving ability remains unchanged in the presence of PEO crystallites because the intercrystalline space is large enough to allow permeation of the hydrocarbons. The presence of PEO crystallites causes a slight permselectivity improvement for the n-butane/methane and the n-butane/ethane gas pairs due to higher solubility selectivity. However, considering the loss of permeability, the presence of impermeable PEO crystallites in the membrane is undesirable for hydrocarbon separation.

## Figures and Tables

**Figure 1 polymers-13-04181-f001:**
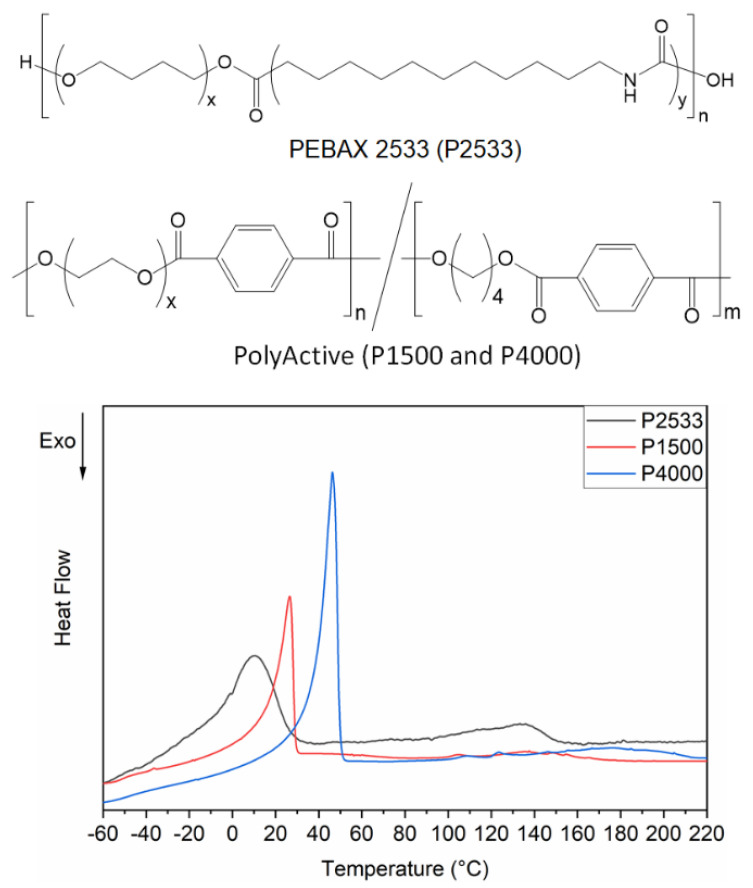
Chemical structures and DSC thermograms (second heating trace) of P2533, P1500 and P4000.

**Figure 2 polymers-13-04181-f002:**
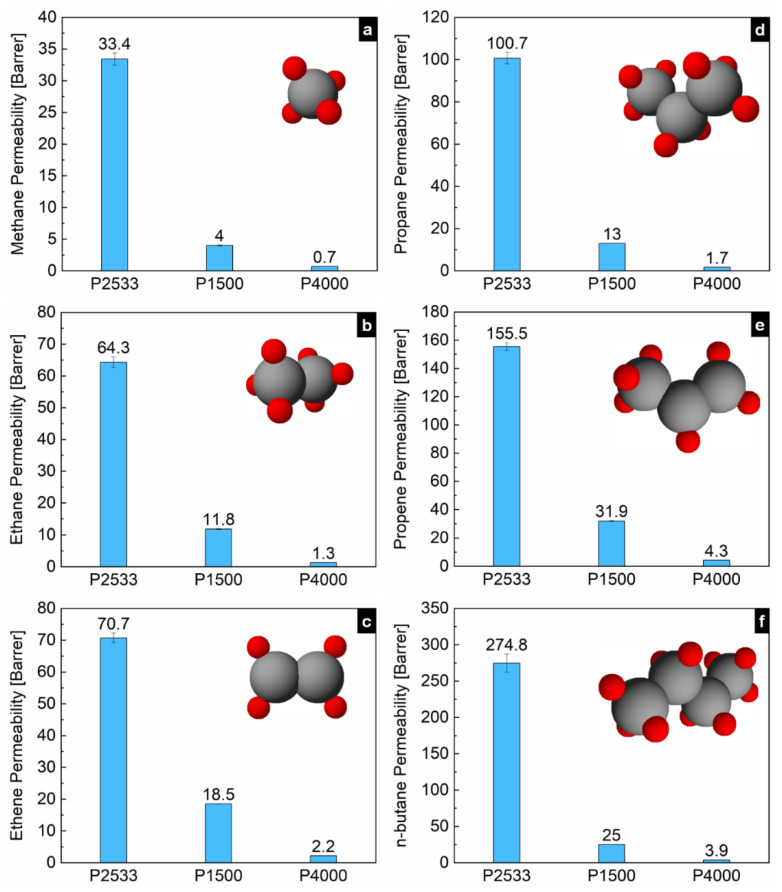
Permeabilities of (**a**) methane, (**b**) ethane, (**c**) ethene, (**d**) propane, (**e**) propene and (**f**) n-butane through P2533, P1500 and P4000.

**Figure 3 polymers-13-04181-f003:**
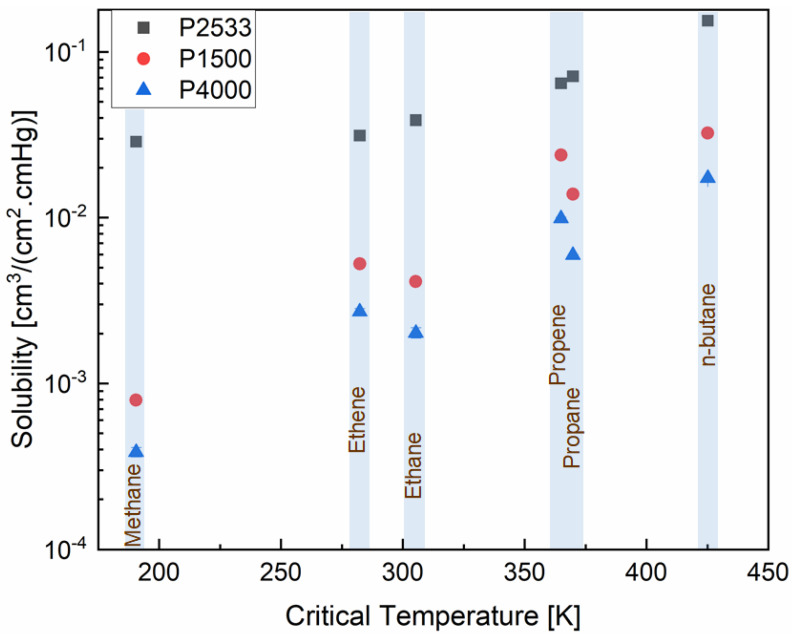
Solubility of the gases in P2533, P1500 and P4000 vs. critical temperature of the gases.

**Figure 4 polymers-13-04181-f004:**
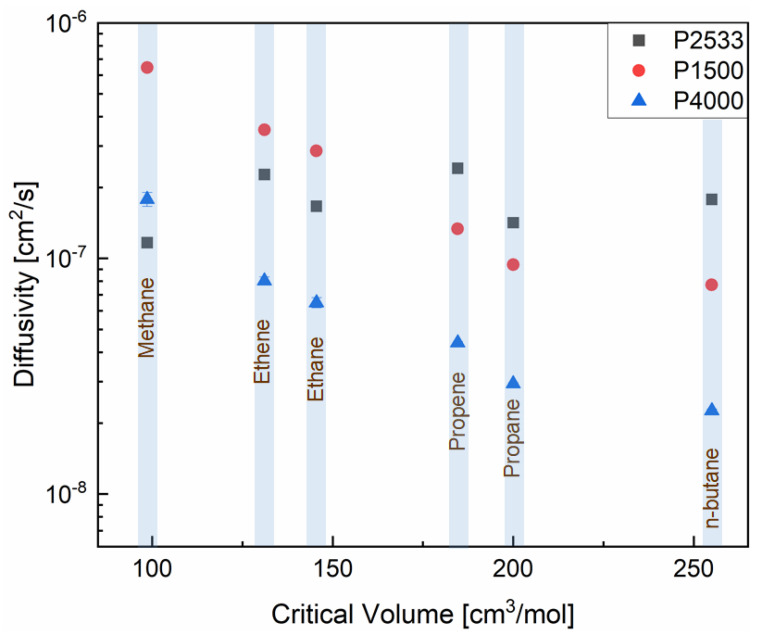
Diffusivity of the gases in P2533, P1500 and P4000 vs. critical volume of the gases.

**Figure 5 polymers-13-04181-f005:**
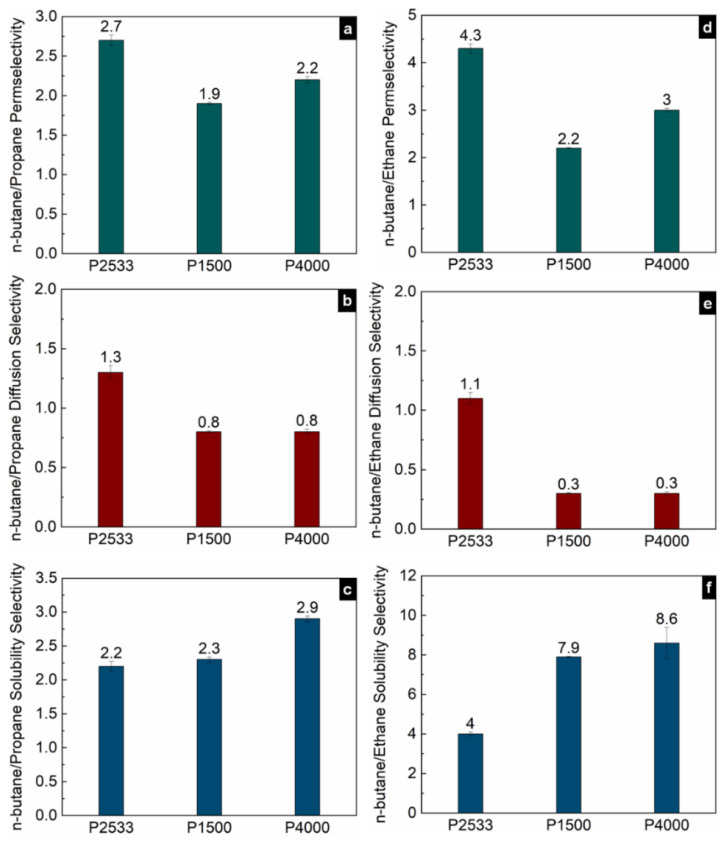
Permselectivities (**a**,**d**), diffusion selectivities (**b**,**e**) and solubility selectivities (**c**,**f**) of n-butane/propane and n-butane/ethane gas pairs in P2533, P1500 and P4000.

**Figure 6 polymers-13-04181-f006:**
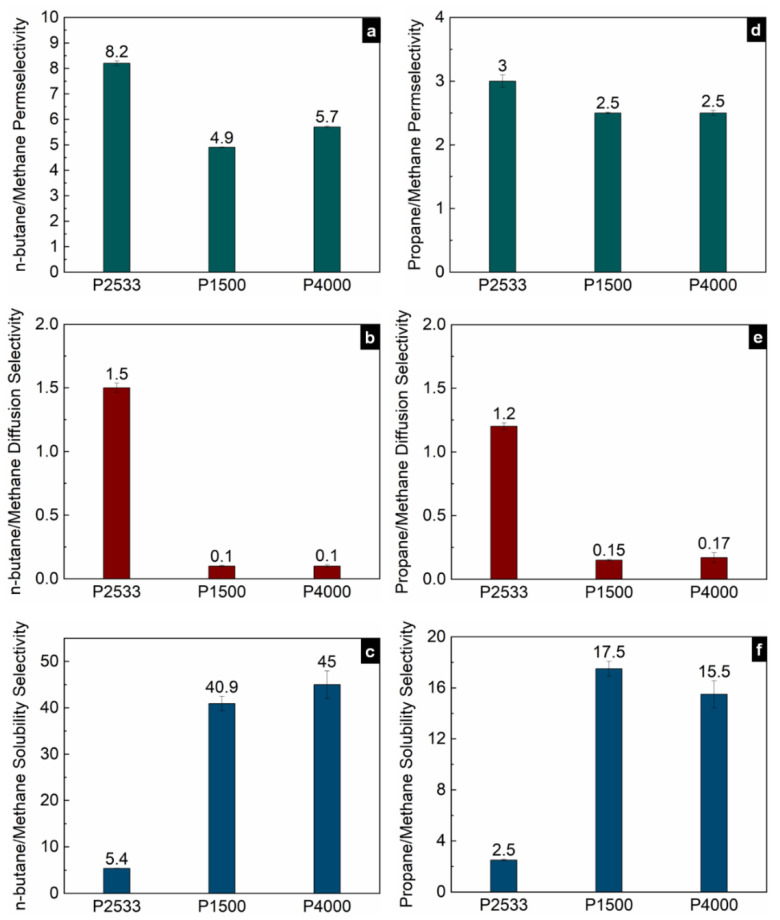
Permselectivities (**a**,**d**), diffusion selectivities (**b**,**e**) and solubility selectivities (**c**,**f**) of n-butane/methane and propane/methane gas pairs in P2533, P1500 and P4000.

**Figure 7 polymers-13-04181-f007:**
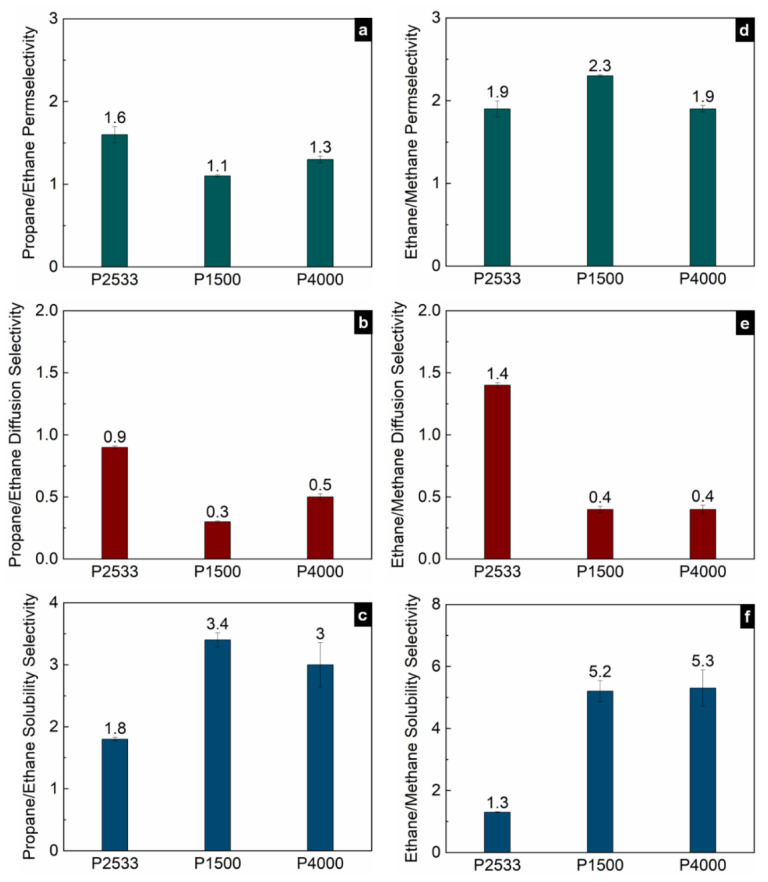
Permselectivities (**a**,**d**), diffusion selectivities (**b**,**e**) and solubility selectivities (**c**,**f**) of propane/ethane and ethane/methane gas pairs in P2533, P1500 and P4000.

**Figure 8 polymers-13-04181-f008:**
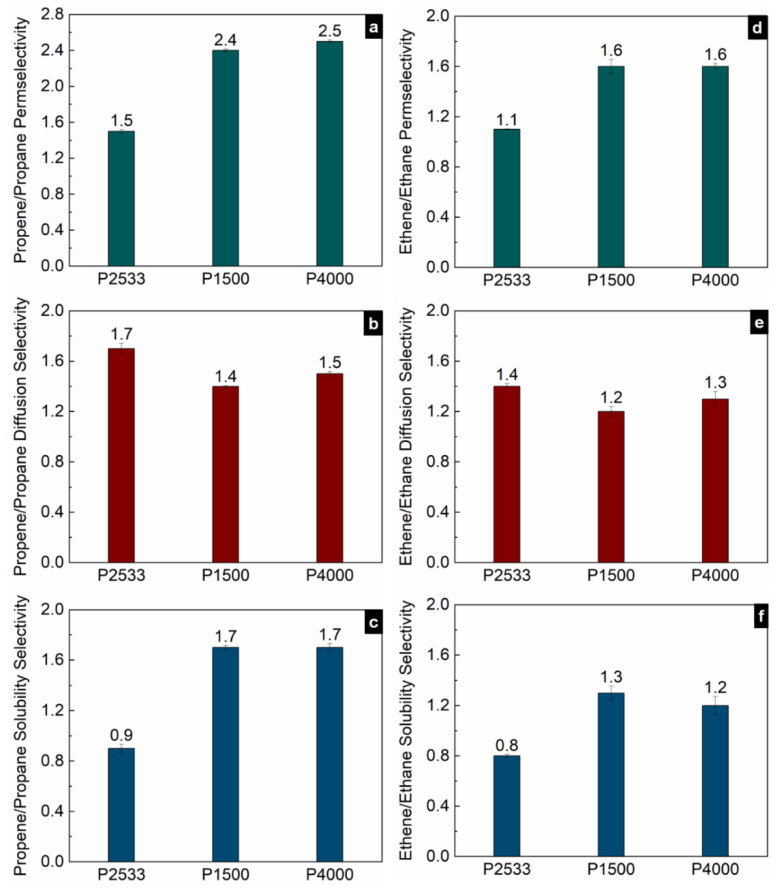
Permselectivities (**a**,**d**), diffusion selectivities (**b**,**e**) and solubility selectivities (**c**,**f**) of propene/propane and ethene/ethane gas pairs in P2533, P1500 and P4000.

## Data Availability

The data presented in this study are available on request from the corresponding author.

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
