# Peer review of "Membrane Separation of Gaseous Hydrocarbons by Semicrystalline Multiblock Copolymers: Role of Cohesive Energy Density and Crystallites of the Polyether Block"

_polymers, 2021, doi:10.3390/polym13234181_

Round 1
Reviewer 1 Report
This manuscript is about the influence of cohesive energy density 3 and crystallites of the polyether block on the separation efficiency of these membranes. In my opinion, after minor revision manuscript can be published.
- line 45: should be carbon dioxide instead of carbondioxide.
- Did the Author consider using the Pebax polymer with PEO segments instead of Pebax with PTMO segments? It has been found that Pebax with PEO segments shows better permeability during the separation of CO2/N2.
- Did Author take into account the free volume in polymer matrix? Free volume also influences on permeability of gases through polymer matrix.
Adding this information will be valuable for future readers. - In my opinion, this issue needs to be discussed more. Lack of definition of cohesive energy density, the equation used for the calculation of cohesive energy density (it is a very difficult issue, not widely used). Second, the relation between the cohesive energy density and solubility should be discussed in more detail.
- line 156 How the cohesive energy density was calculated/determined? There is no reference related to the cohesive energy density. Is it an estimation value? (there are no references)
- line 170 (Hence...) This conclusion is a little overestimated. There are no big differences (solubility parameters). From the literature is well known that rubbery polymers showed high permeability but low selectivity in the separation of olefin/paraffin mixture.
- line 280 What about performing the experiments with P4000 at higher temperatures? If there are no crystallites in the polymer matrix the diffusion coefficient will increase? And subsequently the permeability.
- Did the Author consider testing membrane in the separation of the mixture instead of single, pure gas? The results with the mixture could be different.
Author Response
Response to the comments of Reviewer 1
This manuscript is about the influence of cohesive energy density 3 and crystallites of the polyether block on the separation efficiency of these membranes. In my opinion, after minor revision manuscript can be published.
- line 45: should be carbon dioxide instead of carbondioxide.
Author’s response: Correction has been made in the revised manuscript according to the reviewer’s comment (page 2 of the revised manuscript).
- Did the Author consider using the Pebax polymer with PEO segments instead of Pebax with PTMO segments? It has been found that Pebax with PEO segments shows better permeability during the separation of CO2/N2.
Author’s response: I have been working with Pebax polymers having PEO segments for more than a decade. The Pebax polymers with PEO segments does not show better permeability than Pebax with PTMO during the separation of CO2/N2. The Pebax with PTMO segments show higher permeability. It is well known in the membrane community. The Pebax polymers with PEO segments show better CO2/N2 selectivity than Pebax with PTMO owing to the affinity of the CO2 with ether oxygen. In this study the 3 polymers (1 Pebax and 2 Polyactives) are used. These polymers have similar polyether block content. As the Pebax has PTMO segment and Polyactives has PEO segment it was possible to study the influence of cohesive energy density. If the Pebax would also have PEO segment it would not be possible to study the influence of cohesive energy density.
- Did Author take into account the free volume in polymer matrix? Free volume also influences on permeability of gases through polymer matrix.
Adding this information will be valuable for future readers.
Author’s response: Free volume of the polymer matrix does not only have influence on gas permeability. If there is no free volume in the polymer matrix the gases can’t permeate through the polymer matrix at all. This is a common knowledge in membrane community. According the suggestion of the reviewer the following part has been added in page 7 of the revised manuscript.
“Diffusion of a gas molecule through the polymeric membranes consists of a series of diffusive jumps. In a rubbery polymer the diffusive jumps are facilitated by the formation of transient free volumes. The segments of the rubbery polymers have sufficient energy for chain rotation, translational motion and vibrational motion, which creates transient free volumes and allows the gas molecules to jump from one sight to another. Hence, it is intuitive that cohesive energy density, combined with other factors, has an influence in formation of the transient free volume in rubbery polymers. It is worth mentioning here due to the influence of other factors (e.g. chemical structure of the polymer) a high cohesive energy density does not necessarily translate in low fractional free volume.[29] To examine diffusion, it is equally important to consider the properties of the gas molecules, as the diffusive jumps are function of the size of the gases.”
- In my opinion, this issue needs to be discussed more. Lack of definition of cohesive energy density, the equation used for the calculation of cohesive energy density (it is a very difficult issue, not widely used). Second, the relation between the cohesive energy density and solubility should be discussed in more detail.
Author’s response: According to the suggestion of the reviewer the following part is added in the manuscript (page 4 of the revised manuscript)-
“By definition cohesive energy density is the internal energy of a substance per unit volume. It is a measure of the interaction energy between the molecules of the substance at a fixed temperature. Stronger the interaction between the molecules higher is the cohesive energy density. The square root of the cohesive enthalpy density is the Hildebrand solubility parameter. For small molecules cohesive energy density can be experimentally determine from the energy of vaporization.[29] As the polymers degrade before vaporization, the co-hesive energy density and the Hildebrand solubility parameter for polymers are usually theoretically determined by group contribution method.[30] Efforts have been made to ex-tract cohesive energy density from other parameters e.g. surface tension and thermal ex-pansivity.[31] The cohesive energy density of PEO and PTMO are widely reported in liter-ature. While there are the inherent limitations to determine the absolute values, it is a common knowledge that PEO has higher cohesive energy compared PTMO. [31-32] The attractive forces between the polyether segments stems from the polar ether oxygen. The higher ether oxygen content of PEO leads to a higher cohesive energy density compared to that of PTMO.”
- line 156 How the cohesive energy density was calculated/determined? There is no reference related to the cohesive energy density. Is it an estimation value? (there are no references)
Author’s response: In this paper the absolute value of the cohesive energy density was not calculated/determined. A qualitative comparison has been made. According the suggestion of the reviewer references are added to support the fact that the cohesive energy of PEO is higher than that of PTMO (page 4 of the revised manuscript).
- line 170 (Hence...) This conclusion is a little overestimated. There are no big differences (solubility parameters). From the literature is well known that rubbery polymers showed high permeability but low selectivity in the separation of olefin/paraffin mixture.
Author’s response: The reviewer is right about the fact that rubbery polymers in general have low olefin/paraffin selectivity. But I disagree with him regarding the statement that the conclusion in the reffered line is overestimated. The referred line is copied here for the convenience of the editor.
“Hence, it is evident that the olefinic hydrocarbons have specific affinity towards the polymers which the paraffinic hydrocarbons do not have.”
The solubility values reported in this paper are average of 3 measurements. The higher solubility of olefins then expected from the condensability of the gases is observed in two separate the PEO containing polymers in this work. It is clear from figure 3 that olefins have higher solubility in the PEO containing Polyactives (i.e. P1500 and P4000). Especially the solubility of propene is 1.7 (figure 8c) times higher than propane although the condensability of propene is lower than propane. If the olefins did not have specific affinity towards the PEO containing polymers, propene would have slithly lower solubility than propane. Such clear experimental observation in two different polymers cannot be ignored. This is a clear experimental evidence that olefins have specific affinity towards the PEO containing polymers and the paraffins do not have such affinity.
- line 280 What about performing the experiments with P4000 at higher temperatures? If there are no crystallites in the polymer matrix the diffusion coefficient will increase? And subsequently the permeability.
Author’s response: Ofcourse at higher temperature the PEO crystallites will melt there will be more amorphous PEO. So the diffusion will be higher and the permeability will increase. But the that is not the goal of this paper. In this paper the measurement was done at 33 °C where the PEO segments of P1500 are in an amorphous state and the PEO segments of P4000 are in a semicrystalline state. Therefore it was possible to study the influence of having PEO crystallites in the matrix on gas permeation mechanism.
- Did the Author consider testing membrane in the separation of the mixture instead of single, pure gas? The results with the mixture could be different.
Author’s response: The study of mixed gas is beyond the scope of this manuscript. It is a good suggestion from the reviewer. I plan to perform it in future.

Reviewer 2 Report
The author has analyzed and described the “Membrane separation of gaseous hydrocarbons by semicrystalline multiblock copolymers.” The work is well explained and well written. I recommend its publication after minor revision. Following are the few points that need to be addressed.
- The author has mentioned in the introduction that “For separation of carbon dioxide from light gases, the PEO blocks are expected to be in an amorphous state under the operating condition as the presence of PEO crystallites have a negative impact on both permeability and selectivity.” It is good if the author briefly explains that how PEO crystallites have a negative impact on the permeability and selectivity
- The authors can explain and give some reference to the conclusion that at 33 °C, the polyether blocks of P2533 and P1500 are completely amorphous while that of P4000 is semicrystalline.
- In line 126, the author mentioned “similar amorphous polyether content in P2533 and P1500, how is it similar?
- The solubilities of the ethene and propene should be more compared to the ethane and propane in figure 3 but seem lower, as the author mentioned in a statement at lines 171 &172 “The specific affinity originates from the polar ether oxygen of the polymers and the double-bonded carbons of the gases.”
- Which factors are in different three commercial multiblock copolymers are affecting the permeability, diffusivity, and solubility selectivities for various hydrocarbons?
Author Response
Response to the comments of Reviewer 2
The author has analyzed and described the “Membrane separation of gaseous hydrocarbons by semicrystalline multiblock copolymers.” The work is well explained and well written. I recommend its publication after minor revision. Following are the few points that need to be addressed.
- The author has mentioned in the introduction that “For separation of carbon dioxide from light gases, the PEO blocks are expected to be in an amorphous state under the operating condition as the presence of PEO crystallites have a negative impact on both permeability and selectivity.” It is good if the author briefly explains that how PEO crystallites have a negative impact on the permeability and selectivity
Author’s response: According to the suggestion of the reviewer the referred sentence is explained in briefly in the paper (page 1 and 2 of the revised manuscript).
“For separation of carbon dioxide from light gases the PEO blocks are expected to be in an amorphous state under operating condition as the presence of PEO crystallites have negative impact on both reduces the gas permeability substantially and selectivity. Gas perme-ation through the non porous polymers occurs due to the presence of fractional free volume (i.e. the volume unoccupied by the polymer chains). Gases cannot permeate through the perfectly packed chain folded crystallites as they are too dense. In other words, the PEO crystallites do not have the free volume to allow permeation of the gases which lowers the gas permeability. The crystallinity of the multiblock copolymers have relatively small influence on selectivity. The overall selectivity of the polymers are mostly dictated by the amorphous phase. However, in some studies it has been reported the crystallites contributes to the size sieving ability of the polymer and thereby have negative impact on the selectivity of carbon dioxide over light gases.[15]”
- The authors can explain and give some reference to the conclusion that at 33 °C, the polyether blocks of P2533 and P1500 are completely amorphous while that of P4000 is semicrystalline.
Author’s response: This conclusion was drawn from the DSC thermogram presented in Figure 1 of the paper. So it is based on experimental evidence. The explanation was also already available in the paper (page 4 of the revised manuscript).
“Figure 1 shows the second heating traces of the DSC thermograms of P2533, P1500 and P4000. The microphase separated multiblock copolymer P2533 has two distinct melting endotherm for the PTMO and polyamide 12 block respectively. Two separate melting endotherms are also visible for P1500 and P4000 for the PEO and poly(butylene terephthalate) respectively.[17] It is noteworthy that the onset and endset of the melting endotherms of the PEO block of P1500 and P4000 are significantly different from each other. For P4000 the melting of the PEO block starts at 40 °C and ends at 49 °C. The gas permeation properties through the polymers are investigated at 33 °C. From the DSC thermograms it is evident that at 33 °C the polyether blocks of P2533 and P1500 are completely amorphous while that of P4000 is semicrystalline.”
- In line 126, the author mentioned “similar amorphous polyether content in P2533 and P1500, how is it similar?
Author’s response: This information was already provided at the introduction of the paper (page 2 of the revised manuscript) with proper reference.
“The three commercially available multiblock copolymers PEBAX 2533, PolyActive1500PEGT77PBT23 and PolyActive4000PEGT77PBT23 are denoted as P2533, P1500 and P4000 in this paper respectively. The chemical structure of these polymers are provided in figure 1. P2533 is composed of 80 wt% PTMO and 20 wt% polyamide 12.[1-2, 18] P1500 and P4000 are composed of 77 wt% poly(ethylene glycol)terephthalate and 23% poly(butylene terephthalate).[3, 17]”
- The solubilities of the ethene and propene should be more compared to the ethane and propane in figure 3 but seem lower, as the author mentioned in a statement at lines 171 &172 “The specific affinity originates from the polar ether oxygen of the polymers and the double-bonded carbons of the gases.”
Author’s response: According to the suggestion of the reviewer the following lines are added in the paper. (page 6 of the revised manuscript)
“Considering the condensability of the gases ethene is expected to have a slightly lower solubility in the polymers compared to ethane and propene is expected to have a slightly lower solubility than propane. Figure 3 shows in P1500 and P4000 the solubility of ethene is slightly higher than ethane while that of propene is significantly higher than propane.”
- Which factors are in different three commercial multiblock copolymers are affecting the permeability, diffusivity, and solubility selectivities for various hydrocarbons?
Author’s response: Cohesive energy of the polyether blocks has a significant influence on the permeability, diffusivity, and solubility selectivities for various hydrocarbons while the PEO crystallites have almost on influence. It was already discussed in the paper. (page 13 and 14 of the revised manuscript)
“Figure 5-7 show contribution of diffusion selectivities and solubility selectivities to the permselectivities of paraffinic hydrocarbon pairs. All three polymers selectively allow the permeation of larger paraffinic hydrocarbons of the gas pairs owing to the dominant solubility selectivity over diffusion selectivity. For P2533 the diffusion selectivity of the paraffinic gas pairs are in the range of 0.9 – 1.5. As the diffusion selectivities are almost equal to 1 it is clear that P2533 has almost no size sieving ability and the permselectivity is merely determined by the solubility selectivity. However, that is not the case for P1500 and P4000. Both of these polymers have substantially stronger solubility selectivities for the paraffinic hydrocarbon pairs compared to those of P2533. Especially for the gas pairs having significantly different e.g. n-butane/methane (figure 6c), propane/methane (figure 6f) the solubility selectivities of P1500 and P4000 are 6-8 times higher than those of P2533. However, owing to the cohesive energy density of the PEO blocks as P1500 and P4000 have size sieving ability (i.e. smaller gases diffuse faster) the diffusion selectivity values are far below 1. The counteracting influence of faster diffusivities of smaller gases and higher solubilities of larger gases in P1500 and P4000 reduces the permselectivities of the paraffinic hydrocarbon pairs compared to those of P2533. The permselectivities of n-butane over ethane (figure 5d) and methane (figure 6a) are slightly higher for P4000 compared to those of P1500. It implies the presence of PEO crystallites leads to slightly higher permselectivities of these two gas pairs which originate from solubility selectivities. The difference in permeselectivities and solubility selectivities of P1500 and P4000 for the other paraffinic and olefinic/paraffinic hydrocarbon pairs are rather insignificant. The PEO crystallites significantly reduces the available surface area for the dissolution of the gases. The significantly higher condensability of n-butane over ethane and methane translates into a slight increase of solubility selectivity in presence of the PEO crystallites. But the trend of solubilities (figure 3) and difference of solubility selectivities of all hydro-carbon pairs (figure 5-8) in P1500 and P4000 implies the characteristic thermodynamic properties of the amorphous PEO of these two polymers are similar. The PEO crystallites of P4000 do not impose sufficient stress to alter the thermodynamic property of the amor-phous PEO. The similar trend of diffusivities (figure 4) and the similar diffusion selectivi-ties of all hydrocarbon pairs (figure 5-8) in P1500 and P4000 proves that the PEO crystal-lites also do not alter the size sieving characteristics of the amorphous PEO part of P4000. From an energy consideration the chain immobilization factor, β (equation 6) is a function of the size permeating gases. The similar size sieving ability of P1500 and P4000 implies the chain immobilization factor, β (equation 6) do not play any significant role in the dif-fusion of the hydrocarbons through P4000. The intercrystalline spaces of P4000 are suffi-ciently larger than the critical volume of n-butane (i.e. the largest hydrocarbon used in this study) which makes β rather insignificant for the diffusion of the gases. The crystal-lites of PEO contributes to the tortuosity factor, τ (equation 6) only. As the hydrocarbons have to follow a longer tortuous path due to the presence of the PEO crystallites, the diffu-sivities through P4000 are lower compared to those through P1500.”
